



# Measurement of Light absorbing particles in surface snow of
# central and western Himalayan glaciers: spatial variability,
# radiative impacts, and potential source regions
Chaman Gul[1,2,3,4], Shichang Kang[1,4], Siva Praveen Puppala[2], Xiaokang Wu[5], Cenlin He[6],
Yangyang Xu[5], Inka Koch[1], Sher Muhammad[1], Rajesh Kumar[6], Getachew Dubache[3]

[1]State Key Laboratory of Cryosphere Science, Northwest Institute of Eco-Environment and Resources, Chinese
Academy of Sciences, Lanzhou 73000, China
[2]International Centre for Integrated Mountain Development (ICIMOD), G.P.O. Box 3226, Kathmandu, Nepal
[3]Reading Academy, Nanjing University of Information Sciences and Technology 219 Ningliu Road, Nanjing,
Jiangsu, 210044 China.
[4]University of Chinese Academy of Sciences, Beijing, China
[5]Department of Atmospheric Sciences, Texas A&M University, College Station, TX 77843, USA
[6]Research Applications Laboratory, National Center for Atmospheric Research, Boulder, CO 80301, USA
Correspondence: Siva Praveen Puppala (sivapraveen.puppala@icimod.org);



**Abstract.** We collected surface snow samples from three different glaciers: Yala, Thana, and Sachin in the central
and western Himalayas to understand the spatial variability and radiative impacts of light-absorbing particles. The
Yala and Thana glaciers in Nepal and Bhutan, respectively, were selected to represent the central Himalayas. The
Sachin glacier in Pakistan was selected to represent the western Himalayas. The samples were collected during the
pre-and post-monsoon seasons of the year 2016. The samples were analysed for black carbon (BC) and water-
insoluble organic carbon (OC) through the thermal optical method. The average mass concentrations (BC 2381.39
ng g$^{-1}$; OC 3896.00 ng g$^{-1}$; dust 101.05 µg g$^{-1}$) in the western Himalaya (Sachin glacier) were quite higher compared
to the mass concentrations (BC 357.93 ng g$^{-1}$, OC 903.86 ng g$^{-1}$, dust 21.95 µg g$^{-1}$) at the central Himalaya (Yala
glacier). The difference in mass concentration may be due to the difference in elevation, snow age, local pollution
sources, and difference in meteorological conditions. BC in surface snow was also estimated through WRF-Chem
simulations at the three glacier sites during the sampling periods. Simulations reasonably capture the spatial and
seasonal patterns of the observed BC in snow but with a relatively smaller magnitude. Absolute snow albedo was
estimated through the Snow, Ice, and Aerosol Radiation (SNICAR) model. The absolute snow albedo reduction was
ranging between 0.48 % (Thana glacier during September) to 24 % (Sachin glacier during May) due to BC and 0.13
% (Yala glacier during September) to 5% (Sachin glacier during May) due to dust. The instantaneous radiative
forcing due to BC and dust was estimated in the range of 0 to 96.48 W m$^{-2}$ and 0 to 25 W m$^{-2}$ respectively. The
lowest and highest albedo reduction and radiative forcing were observed in central and western Himalayan glaciers,
respectively. The potential source regions of the deposited pollutants were inferred using WRF-Chem tagged-tracer
simulations. Selected glaciers in the western Himalayas were mostly affected by long-range transport from the
Middle East and Central Asia; however, the central Himalayan glaciers were mainly affected by local and South
Asia emissions (from Nepal, India, and China) especially during the pre-monsoon season. Overall, South Asia and
West Asia were the main contributing source regions of pollutants.
**Keywords:** black carbon; organic carbon; Yala glacier; Thana glacier; Sachin glacier; snow albedo



## 1 Introduction

Black carbon (BC) is a distinct type of carbonaceous material that is formed primarily in flames. BC particles in the atmosphere are generally produced by the incomplete combustion of fossil fuel, biofuel, and biomass. BC is only a minor contributor to aerosol mass but has great climatic interest as a strong absorber of solar radiation (Quinn et al., 2008; Ramanathan and Carmichael, 2008). In addition to warming, BC particles can interact with clouds, changing their microphysical properties, and thus impacting the climate (Wang et al., 2018; Bond et al., 2013; Dong et al., 2021). Besides this, several studies in the past highlighted the role of BC on the cryosphere (Kang et al., 2019, 2020).

The cryosphere is one of the most sensitive indicators of climate change. The temperature rise in cryospheric regions is generally larger than that in other regions on the global scale (Pepin and Lundquist 2008; Kang et al., 2010; You et al., 2021; Huang et al., 2019). BC particles deposit on the glaciers or snow cover surface, decreasing the surface albedo and absorbing more solar radiation (Warren and Brandt, 2008; He et al., 2017) which accelerates snow and ice melt and triggering albedo feedback (Flanner et al., 2009; Hansen and Nazarenko, 2004; Kang et al.,2020). The forcing produced by BC and other light-absorbing particles (LAPs) further affects the regional climate (Flanner et al., 2009; Xu et al., 2016; Ji et al., 2015) leads to complex responses of the Earth climate system (Hansen et al., 1997). The largest climate forcing from BC in the snow is estimated to occur over the Tibetan Plateau (TP) and Himalayas (Flanner et al., 2009; Ji et al., 2015).

Mountain glaciers are the most important freshwater resources to the lives of arid and semi-arid regions (Hock, 2005, 19; Mayer et al., 2006). The great Himalayas is considered as world's largest freshwater reservoir outside the Polar Regions (Immerzeel et al., 2010; Marcovecchio et al., 2021). The economy and lives of millions of people in the region are influenced by the changes in mountain river discharge downstream of the Himalayas (Vaux et al., 2012). Lack of in-situ data, the low resolution of emission inventory, and coarse model resolutions prevent an accurate evaluation of LAPs impacts on snow albedo and radiative forcing. Many glaciers have retreated in the region due to climate warming (Zhang et al., 2009; Kang et al., 2010; Yao et al., 2019), and possibly due to LAP-induced surface darkening (Flanner et al., 2009; Qian et al., 2011; Kang et al., 2019). Glacier retreat in the TP and the Himalayan region has serious consequences because snow and runoff from this region are sources of major rivers in Asia, and the availability of freshwater resources has profound effects on human health and agriculture (Immerzeel et al., 2010). However, it is still large uncertainties for glacier retreat driven predominately by the deposition of BC and other LAPs (Bolch et al., 2012; Kang et al., 2020).

Snow albedo is an important indicator of surface energy budget over the snow-covered area. Small changes in surface snow albedo can have large impacts on surface warming due to the rapid feedbacks involving changes to sublimation, snow morphology, and melt rates (Bond et al., 2013). The concentration of LAPs in surface snow is a major factor that affects snow albedo. BC and other LAPs present in the snow reduce the albedo in the visible portion of the electromagnetic spectrum (e.g., Warren and Wiscombe, 1980, Flanner et al., 2007). Besides the





concentration of pollutants deposited on the surface of the snow, multiple other factors, such as solar zenith angle (SZA), snow grain size, snow shape, snow texture, snow density, and snowpack thickness, can also affect snow albedo (He and Flanner, 2020).  The radiative transfer model used for the albedo has brought a better understanding of snow optical properties in the shortwave spectrum (He and Flanner, 2020). We estimated the spectral snow albedo using the online Snow, Ice, and Aerosol Radiative (SNICAR) model (Flanner and Zender, 2006). The model was originally developed by Flanner et al., 2007, further updated by He et al. (2018) and Dang et al., (2019), and has been widely used in simulating the impacts of LAPs on snow albedos (Qu et al., 2014).

Here we present the mass concentration of BC, water-insoluble organic carbon (OC), and mineral dust in surface snow from the ablation and accumulation zones of selected glaciers, located in three different countries (Nepal, Bhutan, and Pakistan) on the southern slope of the Himalaya. The Yala and Thana glaciers were selected from the central Himalayas, while the Sachin glacier was selected from the western Himalayas. To reasonably compare the results (mass concentrations, and optical and radiative properties) across the central and western Himalayas, samples were collected on similar dates of the same seasons (pre-monsoon and post-monsoon). We investigate the spatial variability of BC, OC, and mineral dust concentrations due to differences in the source region, meteorology, deposition, and post-deposition processes. The measured mass concentrations were compared to regional model simulations. The associated changes in surface snow albedo and radiative forcing (RF) by mineral dust and BC in surface snow were estimated using the SNICAR model. We also aim to identify the potential source regions of pollution reaching sampling sites using tracer-tagged model simulations.

## 2  Study area and meteorology

Samples were collected from the Yala glacier (28º14' N, 85º37' E) in the Langtang valley of Nepal, the Thana glacier (28º01' N, 90º36' E) in the Chamkhar valley of Bhutan, and the Sachin glacier (35º19' N, 74º45' E) in northern Pakistan (Table 1). Monthly mean surface air temperature and precipitation (MERRA-2 reanalysis data) over the selected glaciers were analyzed and compared for the western and central Himalayan glaciers from April 2015 to October 2017 (Table 2). Yala Glacier is a plateau-shaped glacier that has an elevation range between 5160 and 5750 m a.s.l. The length of the Yala glacier is 1.5 km facing the northwest. The glacier is located away from the residential area and is mostly covered by firm/snow especially during the winter season. Details about the metrological condition at the Yala glacier are available in Mukesh et al., (2019) and Gul et al., (2021). Thana glacier is a gentle slope glacier with slight debris cover and an elevation range between 5250 and 5700 m a.s.l. The length of the glacier is about 5 km, facing the southwest. The Thana glacier is mostly covered by fresh snow especially during the winter season. The Sachin glacier has a gentle slope with dense debris cover in its ablation area with an altitude range from 3105 to 4976 m a.s.l. The length of the Sachin glacier is around 8 km facing northeast. In general, the Sachin, glacier is low elevated and relatively debris-covered glaciers as compared to central Himalayan glaciers (Yala and Thana). Precipitation in central Himalayan glaciers (Yala and Thana) was higher than that of the western Himalayan glacier (Sachin) especially from April to October each year (Table 2). Surface air temperature over the Yala and Sachin glacier was higher than that of the Thana glacier. The geographical location of the selected





glaciers and snow sampling locations are shown in Fig. 1. Besides the difference in altitude, latitude, and
meteorology of the selected glaciers in the central and western Himalayas, there is also a difference in the surface
condition shown in supplementary Fig. S1.

**3 Methodologies**

**3.1 Snow sampling and analysis**
Snow samples were collected from the central and western Himalayan glaciers during May, and September 2016.
Samples were taken from the surface of the selected glaciers; however, few snow samples were also collected from
the surrounding nearby areas of the Yala and Sachin glaciers. The snow density was measured with a small density
kit. The snow grain size was measured through a hand lens (25×) with an accuracy of 0.02mm. A detailed
description of the sampling procedure is described in Gul et al., 2021. Quartz filters were used to measure the mass
concentration of BC, OC, and dust in the collected samples. BC and OC present in snow samples were analyzed by a
filter-based thermal-optical analysis method using DRI® Model 2005 (Chow et al., 1993). Filters were analyzed at
the State Key Laboratory of Cryosphere Science, Northwest Institute of Eco-Environment and Resources, Chinese
Academy of Sciences. Before starting the analysis, a piece of the sampled filters was put in an oven for a few
minutes to eliminate the water vapor content and volatile organic compounds. Further detailed information on the
instrument and analysis method can be referred to in earlier studies (Gul et al., 2018, 2021).

**3.2 Estimation of snow albedo reduction and radiative forcing**
The online snow simulation model SNICAR (Flanner et al., 2007, http://snow.engin.umich.edu/) was used to
estimate snow albedo calculation for the collected samples. The model has been used by multiple studies in the past
(e.g., Li et al., 2017;  Gul et al., 2018; Zhang et al., 2018 ). Albedo was simulated based on an hourly SZA at the
sampling site with an averaged mass concentration of BC, dust, and other input parameters such as snow grain size,
snow density, and snow depth from measurements. We computed broadband snow albedo for direct solar incident
radiation under the mid-latitude winter clear sky condition, (Supplementary Table S1). Depending on geographical
location, 10 to 15 SZAs were used (between 0° and 90°) during instantaneous daytime albedo simulation. Albedo
was simulated in four categories: 1- broadband albedo with BC and dust in snow, 2- broadband albedo with BC in
snow only, 3- broadband albedo with dust in snow only, and 4- broadband albedo with the absence of BC and dust
which was considered as a reference albedo. Radiative implications caused by snow darkening due to BC and dust
deposition were investigated using the albedo reduction and the radiative transfer model Santa Barbara DISORT
Atmospheric Radiative Transfer (SBDART) (Ricchiazzi et al., 1998). To evaluate the amount of additional solar
radiations absorbed by the snow in the presence of BC and dust, we estimated the mean solar irradiance and its
characteristics via SBDART, which has been used in the past (Yang et al., 2015). According to the location of the
sampling site, the characteristics of the atmospheric profiles such as water vapor, aerosols, ozone, etc. were set in the
model. RF-based on measured BC and dust concentration in our samples were estimated using the following
equation.



$RF_x = R_{in-short} * \triangle \boldsymbol{\alpha}_x$ --------------- (1)
where, $R_{in-short}$ denotes incident short-wave solar radiation for selected SZA and $\Delta\alpha_x$ denotes the reduction in albedo
due to BC, dust, or both, as simulated by the SNICAR model.

**3.3 Potential source region of pollutants**
Glaciers of the Himalaya Karakoram and Hindukush (HKH) region are located at high altitudes as compared to the
sources of the major pollutants. LAPs including BC and dust can transport from urban areas towards glaciated areas
(e.g., Yasunari et al., 2009; Kang et al., 2019). Multiple approaches, including climate circulation modeling,
combinations of bottom-up inventories, and back air trajectories have been used in the past to determine the possible
source regions of pollution in the HKH region. To identify the potential source region of pollution for the central
and western Himalayan glaciers, the weather research and forecasting (WRF) model coupled with chemistry (WRF-
Chem version 3.9.1.1) (Grell et al., 2005) tagged-tracer simulations for the selected sites.

WRF-Chem simulations were used to estimate BC mass concentration in surface snow and deposition of BC
particles on three selected glaciers (Yala, Thana, and Sachin). We archived the hourly model results for
instantaneous BC deposition and concentration in snow. The horizontal grid spacing of the model was 20 km x 20
km with 35 vertical levels stretching from the surface up to 50 hPa (~20 km). The updated Model for OZone And
Related chemical Tracers (MOZART) was applied for the gas phase chemistry (Knote et al., 2014) while aerosols in
the WRF-Chem were simulated via the Model for Simulating Aerosol Interactions and Chemistry (MOSAIC)
(Zaveri et al., 2008). We use the Global Data Assimilation System (GDAS) from the National Center for
Environmental Prediction (NCEP) for the meteorological initial and boundary conditions. We used the Fire
Inventory from NCAR (FINN), the EDGAR-HTAP, and MEGAN (Model of Emissions of Gases and Aerosols from
Nature) for biomass burning emissions, anthropogenic emissions, and for online biogenic emissions, respectively.
For chemical boundary conditions, we used the NCAR global CAM-Chem simulation dataset
(https://rda.ucar.edu/datasets/ds313.7/). Key meteorological variables such as winds, temperature, and water vapor
above the planetary boundary layer (PBL) were nudged every 6 hours towards the NCEP GDAS reanalysis fields to
reduce temporal error growth in meteorological variables. We used the Community Land Model (CLM) scheme for
the land component in WRF-Chem. The CLM model can simulate BC concentration in snowpack and its effects on
snow albedo (Flanner et al., 2007). We used online coupled BC deposition fluxes from the atmosphere component
of WRF-Chem with the CLM model following Zhao et al. (2014). We also implemented a tagged-tracer method
(Kumar et al., 2015) to track anthropogenic BC emissions from 10 different Asian countries surrounding the TP
areas, as well as BC emissions from Asian biomass burning and the domain boundary (i.e., areas outside Asia). The
tracked 10 anthropogenic emission source regions include China, India, Nepal, Pakistan, Afghanistan, Bhutan,
Bangladesh, Myanmar, Southeast Asia, and the rest of Asia. The aim of the model simulation was to estimate the
BC mass concentration in surface snow, deposition of BC particles, and the source contribution to BC deposition on
snow.





## 4. Results and discussions

### 4.1 Concentrations of light-absorbing particles in surface snow

The average mass concentration of LAPs in surface snow of the Yala glacier was 357.93 ng g$^{-1}$ for BC, 903.86 ng g$^{-1}$ for OC, and 21.95 µg g$^{-1}$ for dust in spring (May) and was relatively lower concentrations of 68.97 ng g$^{-1}$ for BC, 177.50 ng g$^{-1}$ for OC and 4.3 ng g$^{-1}$ for dust during autumn (September). These mass concentrations of BC and OC in surface snow were comparable to the study result conducted on the Yala glacier in May 2017 (Gul et al., 2021). High LAP concentration in the pre-monsoon is due to a seasonally high Indian subcontinent emission (Kang et al., 2019; Gul et al., 2021); Lau et al., (2010) also confirmed that aerosols from biofuel and biomass burning rapidly build up over Indo-Gangetic Plains (IGP) and East Asia during pre-monsoon season and move towards the study site. The average surface concentrations of BC, OC, and dust in the Thana glacier samples during the autumn season were 39.39 ng g$^{-1}$, 115ng g$^{-1}$ and 34.63µg g$^{-1}$, respectively. Possible reasons for the lowest concentration at the Thana glacier may be due to the relatively high elevation of sampling location and relatively fresh snow. A strong effect of LAPs (BC and dust) has been observed at lower elevations in comparison to higher elevations (Li et al., 2017). The average concentration of BC, OC, and dust measured in the selected western Himalayan glacier (Sachin) during May were 2381.39 ng g$^{-1}$, 3896 ng g$^{-1}$ and 101 µg g$^{-1}$, respectively, and were relatively higher during October with values of 5314 ng g$^{-1}$ for BC, and 546 µg g$^{-1}$ for dust (Gul et al., 2018). The observed average mass concentrations in the western Himalayas were higher than those in the central Himalayas. The BC mass concentration difference might be due to the difference in snow type, precipitation rate, and local emission, the elevation of sampling sites, meteorology, and BC deposition over the glacier surfaces. Post dry deposition of LAPs over the surface of the snow was an important factor. Snow samples collected from the western side of the Himalayas were aged as compared to the central side; post-deposition ion (or enrichment) of LAPs over the snow surface increased the concentration in the snow (Kang et al., 2019). The majority of the samples from the western Himalayan side were from ablation zones of the glacier, where concentrations of LAPs are higher as compared to the accumulation zone of the glacier. Li et al., (2017) showed a strong negative relationship between the elevation of glacier sampling locations and the concentration of LAPs. Therefore strong melting of surface snow and ice in the glacier ablation zone could lead to BC enrichment which causes high BC concentrations (Li et al., 2017). In the case of western Himalayan glaciers sites, snow samples were collected long after the snowfall and the concentration of pollutants would also have increased in the surface snow due to dry deposition. The surface concentrations of the individual samples collected from the Yala, Thana, and Sachin glaciers during May and September 2016 are shown in Fig. 2, and Table S2. BC and OC concentration on our selected glaciers with a comparison to other glaciers of TP and the surrounding region are shown in Fig. 4 and Table S3. It was observed that the concentration of BC, OC, and dust in the central Himalayan glaciers (Yala and Thana) were comparable to other reported results. In the past, similarly high concentrations were reported in the region (Xu et al., 2012) such as Tien Shan Mountains (Li et al., 2016), Northeast of the TP (Wang et al., 2016), Northern China (Zhang et al., 2016) Southeastern TP, western Tien Shan and Central Asia (Zhang et al., 2017).



The yellow boxes of Fig. 2 show the WRF-Chem modeled BC concentrations in surface snow at the three
measurement glacier sites during the measurement periods. Compared to the observations red boxes in Fig. 2, model
results reasonably capture the spatial and seasonal patterns and variables of the observed BC in the snow with a
relatively smaller magnitude. The modeled variation at the Sachin site during the October sampling periods is much
larger than the observations (Gul et al., 208). The discrepancies between model results and observations are due to
model uncertainties from (1) the relatively coarse grid spacing that may not capture the transport over the complex
TP terrain, (2) the underestimated anthropogenic emissions that are not representative for the measurement periods,
and (3) deficiencies in model physical parameterizations that affects BC transport and deposition. We also note that
the observed variation at each site shown in Fig. 2 includes both the temporal and subgrid variabilities derived from
multiple sampling locations surrounding each site (Fig. 1). In contrast, all the measurement locations at each
particular glacier site are located within a single model grid. As a result, the model is unable to resolve this subgrid
information and hence only includes the temporal variability for each selected site.

**4.2 Surface snow albedo and radiative forcing**
The minimum daytimes absolute albedo reduction due to combined BC and dust,  BC only and dust only were in the
range (1.03-13.44%), (0.48-12.42%) and (0.12-2.12%), respectively. The maximum daytime albedo reduction due to
combined BC and dust,  BC only and dust only was in the range (1.98-24.97%), (1.05-24%), and (0.25-4.8%)
respectively. The lowest and highest contributions in albedo reduction were observed in the central Himalayas
(September) and the western Himalayas (May) respectively. Snow albedo reduction (%) derived from samples
collected from the Yala glacier (during May 2016) and the Thana glacier (during September 2016) were in the range
of (0.13-3.82%) and (0.90-1.99%), respectively. A significant difference in daytime albedo reduction between the
western and central Himalayas was mainly due to the difference in mass concentrations of pollutants and snow age.
The pollutant concentrations in the western Himalayan samples (Sachin glacier) were higher, resulting in higher
albedo reduction as compared to the central Himalayan (Yala and Thana glaciers) samples. The average elevation
difference between central and western sampling sites was greater than 1000 meters, where a high concentration of
pollution is expected at the low elevated glacier of the western side as compared to the central side of the Himalaya.
Snow samples collected on the central side of the Himalayas (Yala glacier) were much fresher as compared to the
samples collected from the western side (Sachin glacier). Dust and other pollutants were visible over the surface of
the Sachin glacier (Fig. S1). Aged snow had increased density, enlarged grain size, and increased concentration of
BC and dust particles due to dry deposition on the snow surface. In the case of all sampling sites impact of BC on
snow albedo reduction was greater than the impact of dust except the Thana glacier where the impact of dust was
higher than that of BC (Fig. 4a). This may be due to a different dust type in Thana samples. Daytime snow albedo
reductions (%) due to BC only, dust only, and both BC and dust are given in Fig. 4a.

The daytime instantaneous RF (W m$^{-2}$) ranged from (0.076 to 39.65) for the Yala glacier during May 2016, 0.006 to
18.26 for the Yala glacier during September 2016, 0.0 to 11.48 for the Thana glacier during September 2016, and
0.03 to 96.48 for the Sachin glacier during May 2016. RF for the western Himalayas (Sachin glacier) was quite high



as compared to the central Himalayan glaciers (Yala and Thana glaciers). The radiative effect on the Sachin glacier
was much more than that of other selected glaciers mainly due to low albedo and increased temperature. Zhang et al.
(2017) reported that a reduction in albedo by 9 to 64 % can increase the instantaneous RF by as much as 24.05–
323.18 W m$^{-2}$. In the case of all sampling sites impact of BC on RF was greater than the impact of dust except the
Thana glacier where the impact of dust was higher than that of BC (Fig. 4b). Therefore, BC can be a major
responsible pollutant in the snow to reduce albedo and increase warming in the selected glaciers. BC was the
dominant factor in snow melting in the Yala and Sachin glaciers; however, dust was the dominant factor in Thana
glacier samples. According to (Kaspari et al., 2014), RF caused by mineral dust was greater than that of dust. The
BC and dust had low importance for RF in fresh snow (central Himalaya - Thana glacier) as compared to aged snow
(western Himalaya - Sachin glacier). In the northern TP, BC played important role in RF (Li et al., 2016a), while in
the central TP and Himalayas dust was more important than BC (Kaspari et al., 2014). The average instantaneous
RF caused by the combined contribution of BC and dust (BC + dust), only BC, and only dust is shown in Fig. 4b as
a function of surface snow types. Variation in the RF and albedo change for a particular pollutant type was due to
variation in SZA.

**5 Potential source regions of pollutants**

Figure 5 shows the contributions of different BC emission sources to the BC in snow from WRF-Chem tagged-
tracer simulations. For the Yala site, it is dominated (>50%) by anthropogenic emissions from India and Nepal for
both May and October, while the biomass burning contribution (>20%) increases largely in May primarily due to the
spring burning activities in northern India (Kumar et al., 2011). In September, China's contribution also increases to
>20% at Yala. For the Thana site, it is dominated (>60%) by anthropogenic emissions from China and India in
September, while anthropogenic emissions from Bhutan and Myanmar also contribute about 10%, respectively. The
Sachin site is predominantly affected by anthropogenic emissions from India and Pakistan (total contribution
>80%), while the spring biomass burning only contributes to ~10% in May. Overall, the source contributions show
large variation depending on the site locations and sampling seasons, but with a consistent India contribution of 20-
40% across all the sites and seasons.

**6 Discussion on uncertainty in measurements, albedo, and potential source identification of pollutants**

The possible uncertainties in the present research were related to measurements, sampling, analysis, albedo, and RF
estimation. A sampling at remote rural sites, sample preservation, filtration, and transport can modify the results if
proper standard protocols were not followed. During laboratory analysis via thermal optical techniques, several
uncertainties may be related to separating OC from BC in the sample (Gul et al., 2021). The level of generated
uncertainty depended on temperature protocol, sample type (residential cookstoves, diesel exhaust, rural aerosols,
and urban aerosols), the amount of dust loading on the filter, and the analysis method. The overall accuracy in the
measurement of OC, BC, and total carbon concentrations was estimated considering the mass contributions from
field blanks and the analytical accuracy of concentration measurements. The uncertainty of the OC and BC mass
concentrations was extracted through the standard deviation of the field blanks (li et al., 2021). OC in snow can



produce minor warming (Yasunari et al., 2015), but in this research albedo reduction from OC was not quantified. In
albedo simulation and RF estimations, snow grain size and texture can produce large uncertainty. We
measured/considered the physical grain size in this research which is not the same as the effects than optical grain
size. Optical grain size defines the amount of solar radiation absorbed/scattered by the snow. We assumed a
spherical shape for the snow grains which may affect the results because the albedo of non-spherical grains is higher
than the albedo of spherical grains (Dang et al., 2016; He et al., 2018). The contribution of pollutants generated from
local sources can be important (e.g., Li et al., 2021), which however was not included in the global emission
inventories; we were unable to capture emissions at the local scale. Therefore contributions of local sources may be
underestimated by coarse-resolution models. High-resolution models and emission inventories at the local scale are
required to capture local emissions.

**7 Conclusions**
The average mass concentration of LAPs in the samples collected from the Sachin, Yala, and Thana glaciers were in
the range (835.324 ng g$^{-1}$ to 3545.35 ng g$^{-1}$ for BC and 35.24 µg g$^{-1}$ to 253.52 µg g$^{-1}$ for Dust), (23.16 ng g$^{-1}$ to
2529 ng g$^{-1}$for BC and 1.5 µg g$^{-1}$ to 196.5 µg g$^{-1}$for Dust), and (21 ng g$^{-1}$ to 127 ng g$^{-1}$ for BC and 1.5 µg g$^{-1}$ to 67
µg g$^{-1}$for Dust) respectively.  Overall the concentrations of BC and dust were varied from 21 ng g$^{-1}$ and 1.5 µg g$^{-1}$ in
fresh snow to 3545 ng g$^{-1}$ and 253 µg g$^{-1}$ in the aged snow, respectively. Mass concentrations of BC, OC, and dust in
the samples collected from the western Himalayas was much higher than the average concentration in the central
Himalayas mainly due to difference in snow age, elevation, and meteorology. The accumulation area of glaciers
(e.g. ice cores and snow pits), where enrichment influences are less marked and measured values are likely to be
lower, and high elevation areas, where deposition of pollutants is expected to be lower. Pollutant concentrations
were likely underestimated in the earlier studies, particularly when there was strong surface melting. Dust and other
pollutants were visible on aged snow surfaces in the western Himalaya, indicating considerable enrichment during
snow aging. WRF-Chem modeled BC concentrations in surface snow were almost similar to the observed BC in the
snow with a relatively smaller magnitude.

Based on observed pollutants, snow albedo reduction (%) in the central Himalayas was in the range of (0.48-3.6%
for BC) and (0.13-1.99% for Dust), much lower than that of the western Himalayas. BC was the major component
responsible for the albedo reduction, and the dust had little effect except in the Thana glacier. In case the of the
Thana glacier, the impact of dust was higher than that of BC. The daytime instantaneous radiative forcing (W m$^{-2}$)
ranged from 0.076 to 39.65 (Yala glacier during May 2016), 0.006 to 18.26 (Yala glacier during September 2016),
0.0 to 11.48 (Thana glacier during September 2016), 0.03 to 96.48 (Sachin glacier during May 2016). The average
albedo reduction due to the combined effect of dust and BC at the western Himalayan side (Sachin glacier) was
0.372 which was˜15 times higher than that of the central Himalayan side (Yala glacier). Similarly, the radiative
forcing in the western Himalayas was ˜6 times higher than that of the central Himalayan side. Observation showed
that the potential source regions of pollutants for the western and central Himalayas were different. Western
Himalayan glaciers were mostly affected by long-range transport via the westerlies; however central Himalayan





glaciers were affected by relatively local winds from Nepal, Bhutan, India, and China. For the western Himalayan
glaciers, the emissions from central Asian and South Asian countries (Particularly Pakistan and India) are more
important source regions.

**Acknowledgment**
This study was supported by the National Natural Science Foundation of China (41630754), the Chinese Academy
of Sciences (XDA20040501, QYZDJ-SSW-DQC039), and the State Key Laboratory of Cryosphere Science
(SKLCS-ZZ-2021). This study was also partially supported by the core funds of ICIMOD contributed by the
governments Afghanistan, Australia, Austria, Bangladesh, Bhutan, China, India, Myanmar, Nepal, Norway,
Pakistan, Sweden, and Switzerland. We thank Arnico Panday, Lubna ayaz and Aditi Mukherji for their useful
comments and guidance. We are also grateful to the staff of the National Centre for Hydrology and Meteorology in
Bhutan for organizing the Thana Glacier expedition in 2016. We would like to acknowledge high-performance
computing support from Cheyenne provided by NCAR's Computational and Information Systems Laboratory,
sponsored by the National Science Foundation. NCAR is operated by the University Corporation for Atmospheric
Research under the sponsorship of the National Science Foundation.

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



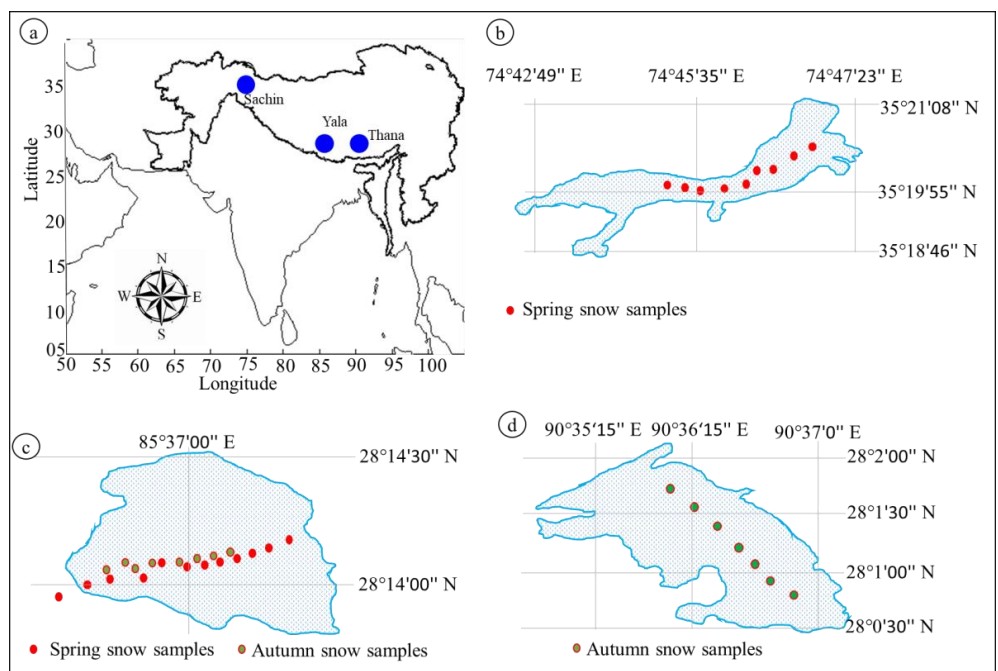


**Fig. 1**. **Study area map (a) locations of selected glaciers in Himalaya Karakoram and Hindu Kush region (b) Sachin glacier in Pakistan (c) Yala glacier in Nepal (d) Thana glacier in Bhutan**


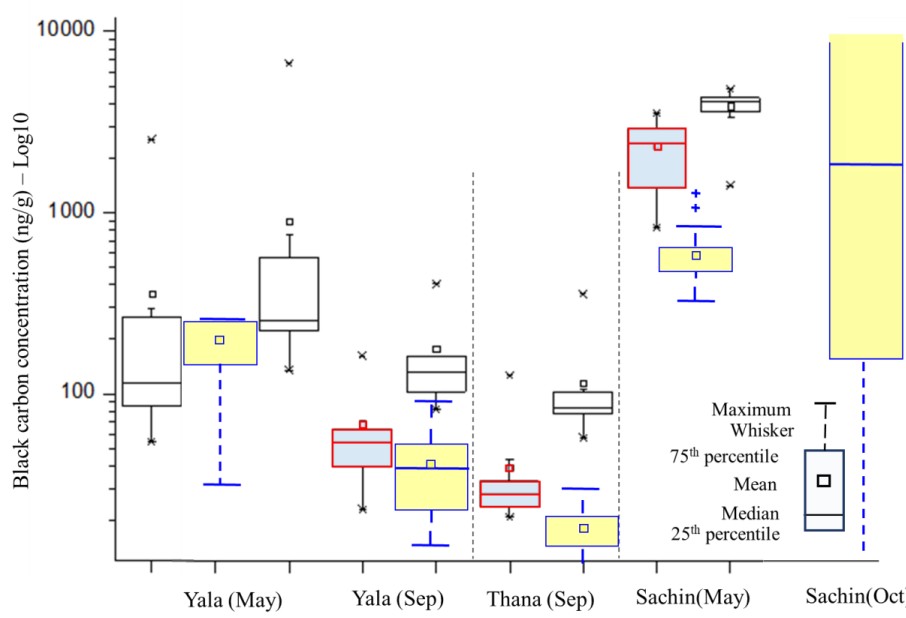


**Fig. 2.  Whisker plots of black carbon (red box) and organic carbon (black box) concentrations (ng/g) in snow samples**
**collected from three different glaciers in spring and autumn 2016.  The yellow boxes are representing BC content in**
**surface snow from WRF-Chem simulations.**



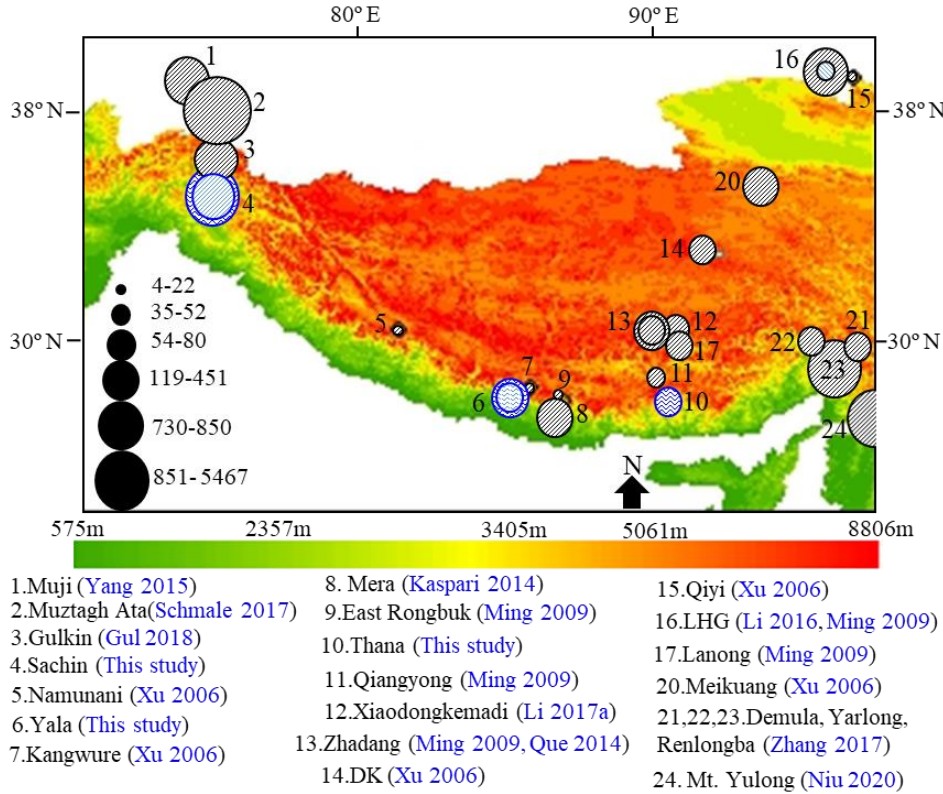

1.Muji (Yang 2015)
2.Muztagh Ata(Schmale 2017)
3.Gulkin (Gul 2018)
4.Sachin (This study)
5.Namunani (Xu 2006)
6.Yala (This study)
7.Kangwure (Xu 2006)

8. Mera (Kaspari 2014)
9.East Rongbuk (Ming 2009)
10.Thana (This study)
11.Qiangyong (Ming 2009)
12.Xiaodongkemadi (Li 2017a)
13.Zhadang (Ming 2009, Que 2014)
14.DK (Xu 2006)

15.Qiyi (Xu 2006)
16.LHG (Li 2016, Ming 2009)
17.Lanong (Ming 2009)
20.Meikuang (Xu 2006)
21,22,23.Demula, Yarlong,
Renlongba (Zhang 2017)
24. Mt. Yulong (Niu 2020)

**Fig. 3. Black carbon concentrations (ng/g) in snow/ice samples in Himalayan, Karakoram and Tibetan Plateau in previous studies (black circles) and this study (blue circles).**



**Fig. 4. (a) Snow albedo reduction due to black carbon, dust and combined (black carbon and dust) during day time for a range of solar zenith angles. (b) Average instantaneous radiative forcing based on albedo reduction values during day time.**





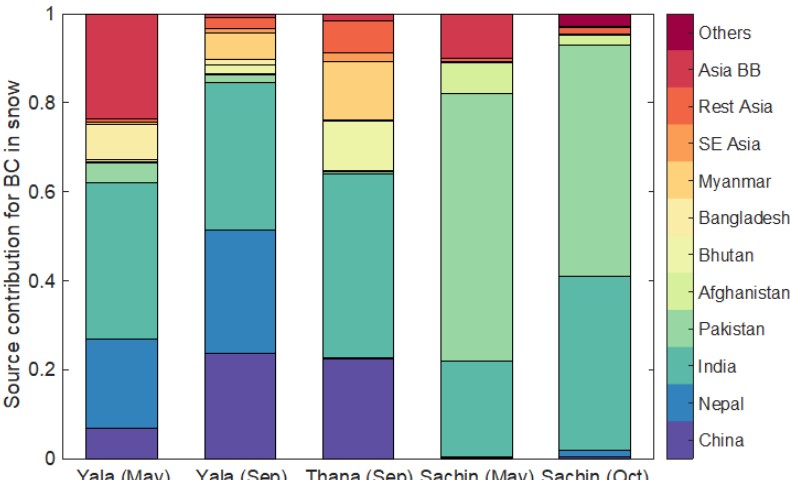

**551**

**552** **Fig. 5. Source contributions to BC content in surface snow from WRF-Chem simulations at the three measurement**
**553** **glacier sites during the measurement periods. Source regions include anthropogenic emissions from China, India, Nepal,**
**554** **Pakistan, Afghanistan, Bhutan, Bangladesh, Myanmar, Southeast (SE) Asia, and the rest of Asia, as well as Asian biomass**
**555** **burning (BB) and BC transported from areas outside the study domain (Others).**

**556**

**557**
**558**
**559** **Table 1. Snow sampling time and locations from selected glaciers**

| Glacier | Lat/Long | Sampling date | Average elevation | Himalayas |
|---|---|---|---|---|
| Yala (Nepal) | 28° 14' 12.25"N, 85° 37' 04.24"E | 4th - 7th May 2016 | 4950 meters | Central |
| Yala (Nepal) | 28° 14' 12.25"N, 85° 37' 04.24"E | 29th September 2016 | 4950 meters | Central |
| Thana (Bhutan) | 28° 01' 22.23"N, 90° 36' 28.72"E | 15th September 2016 | 5400 meters | Central |
| Sachin (Pakistan) | 35° 19' 55"N, 74° 45' 35"E | 15th May 2016 | 3230 meters | Western |

**560**

**561** **Table 2. Comparison of BC mass concentration, albedo reduction, radiative forcing and potential source regions of**
**562** **pollutants for central and western Himalayan glaciers during the study period**

| | Central Himalaya min – max (average) | Western Himalaya min – max (average) | Time period |
|---|---|---|---|
| Monthly mean temperature (°C) | 2.05-14.36(10.35) Yala -9.11- 5.68(0.23) Thana | -10.78 - 14.63 (3.57) Sachin | Apr 2015-Oct 2017 |
| Monthly mean precipitation (mm day$^{-1}$) | 0.04536 - 41.472 Yala 1.0195 - 50.112 Thana | 0.1546 - 5.866 (Sachin) | Apr 2015-Oct 2017 |
| Elevation of sampling location (meters) | 4580-5675 (5127) | 3134-3957(3545) | |
| Observed BC in surface snow (ng g$^{-1}$) | 21 – 2529 (~350) | 835 – 3545 (~2300) | 2016 |
| Albedo reduction (%) due to BC particles in snow | 0.13-3.82 | 12.00-24.00 | 2016 |
| Instantaneous radiative forcing | 0.0-39.65 | 0.03 to 96.48 | 2016 |





| | | |
|---|---|---|
| (W m$^{-2}$) due to BC particles | | |
| Potential source regions of pollutants | | |
| 3. WRF-Chem simulations | For the Yala site, it is dominated (>50%) by anthropogenic emissions from India and Nepal for both May and October, while the biomass burning contribution (>20%) increases largely in May.<br>For the Thana site, it is dominated (>60%) by anthropogenic emissions from China and India in September, while anthropogenic emissions from Bhutan and Myanmar also contribute about 10%, respectively. | For the Sachin site, it is predominantly affected by anthropogenic emissions from India and Pakistan (total contribution >80%), while the spring biomass burning only contributes to ~10% in May. |

563
564