# Peer review of "Measurement of light absorbing particles in surface snow of"

_Atmospheric Chemistry and Physics, 2021_

## Author Response (AR2)

**Journal:** Atmospheric Chemistry and Physics
**Manuscript ID:** ACP-2021-935
**MS type:** Research article
**Title:** "Measurement of light absorbing particles in surface snow of central and western Himalayan glaciers: spatial variability, radiative impacts, and potential source regions".

**A detailed response to reviewer#1 comments**

We thank reviewer 1 for his/her constructive comments and suggestions, which help to improve the manuscript.
On behalf of all the authors, we would like to convey our gratitude to the editor and the reviewers for considering the present work. We have tried our best to incorporate the individual comments. The reviewer's comments are in black, the author's replies are in blue and the modified/inserted text in the revised manuscript is in orange. Below is a point-by-point response to the comments.

**General comments**

The manuscript seems to be closely related to a 2021 EGU presentation (https://meetingorganizer.copernicus.org/EGU21/EGU21-8515.html), and a 2021 paper in Environmental Pollution (https://www.sciencedirect.com/science/article/pii/S0269749121001226?via%3Dihub ), by many of the same authors as the submitted manuscript. However, the latter is focused on results from 2017, whereas the present results focus on 2016. Overall, the methodology is sound; in particular, analysis of black carbon and organic carbon in snow is based on Thermal-Optical Analysis, which is widely employed to distinguish organic carbon from elemental carbon in atmospheric aerosols. A novel aspect of the work is the integration of WRF-Chem to compare to observations of impurities in snow.

Response to the general comments:
The published manuscript in the journal of Environmental pollution, which was also presented in an EGU meeting was mainly discussed year-long observations of atmospheric black carbon mass concentration at a high-altitude site located in Nepal. Few snow samples from the Yala glacier (Nepal) were also collected (May 2017) and were analysed for black carbon and organic carbon, via thermal-optical analysis. However, the current manuscript is discussing the spatial variability of light absorbing particles in surface snow from three glaciers in three different countries (Nepal, Bhutan, and Pakistan, for the year 2016 as shown in Table 1) and their radiative impacts, and possible source regions.

**Table 1. Snow sampling time and locations from selected glaciers**

| Glacier | Lat/Long | Sampling date | Average elevation | Himalayas |
|---|---|---|---|---|
| Yala (Nepal) | 28° 14' 12.25"N, 85° 37' 04.24"E | 4th - 7th May 2016 | 4950 meters | Central |
| Yala (Nepal) | 28° 14' 12.25"N, 85° 37' 04.24"E | 29th September 2016 | 4950 meters | Central |
| Thana (Bhutan) | 28° 01' 22.23"N, 90° 36' 28.72"E | 15th September 2016 | 5400 meters | Central |
| Sachin (Pakistan) | 35° 19' 55"N, 74° 45' 35"E | 15th May 2016 | 3230 meters | Western |

**Specific comments addressing individual scientific questions/issues (section)**

Comment#1
Introduction line 87: I am not sure what "snow shape" is. And I think "snow texture" might mean "snow surface texture". For both, it would help if the authors clarified the scale of these features: millimeters? meters?

Response #1
Thanks for pointing it out. In the revised text we have replaced snow shape and snow texture with snow grain shape and snow surface texture respectively.

The snow grain size was measured in millimeters through a hand lens (25×) with an accuracy of 0.02mm (section 3.1, lines 137, 138). The snow grains are not perfectly spherical and there is a portion of grains with either spheroid or aggregating shapes (Discussed in He et al., 2018, ACP). The albedo of an opaque snowpack with equidimensional nonspherical snow grains is higher than that with spherical snow grains (Dang et al., 2016).

We did not measure the snow surface texture in this study. In the past texture of the buried surface-hoar layer was measured with the help of in situ microphotography (Jamieson and Schweizer, 2000).

The revised text (line 87) is given below
"Besides the concentration of pollutants deposited on the surface of the snow, multiple other factors, such as solar zenith angle (SZA), snow grain size, snow grain shape, snow surface texture, snow density, and snowpack thickness, can also affect snow albedo (He and Flanner, 2020)."

Comment#2
I find the description of the snow sampling to be inadequate. On lines 133-134, the manuscript refers the reader to a 2021 paper by Gul et al for details, but that paper describes sampling in a different year (2017), with specific references to dates that obviously can't apply to the present work, e.g., Before the commencement of snow sampling on May 1, 2017, there was fresh snowfall around the study site. The mean snow thickness of fresh snow was around 15–18 cm and we collected samples from the top 7–10 cm layer. Critical aspects of the sampling protocol employed in this manuscript are therefore unclear to me: were the samples taken in 2016 (this work) also mainly of fresh snowfall? How thick was the fresh snow? At what depths were samples taken? At altitudes above 5000 meters, it will be obvious to some readers that the area lies above the tree line (which is a common source of debris at lower altitudes), but it would be useful to state that; a photograph of the sampling might also be useful.

Response #2
As suggested, we have modified the description of the snow sampling section accordingly (lines 130-145) and the same is given below. We have cited another relevant paper instead of Gul et al., 2021. Photograph of selected glaciers is added in supplementary Fig. S1.
Surface snow samples were collected from the central and western Himalayan glaciers during May, and September 2016. Samples were taken from the ablation and accumulation zones of the selected glaciers; however, a few snow samples were also collected from the surrounding nearby areas of the Yala and Sachin glaciers. Sachin glacier's samples were relatively aged snow and had less snow thickness as compared to the samples collected from Thana and Yala glaciers (supplementary Fig. S1). At each sampling location, Whirl-Pak bags were used to collect samples from the upper 0-10 cm of depth (approximately 2 L, unmelted). The samples were kept frozen until they were melted and

filtered in through the quartz filters near the sampling site. The snow density was measured with a small density kit. The snow grain size was measured through a hand lens (25×) with an accuracy of 0.02mm. The same sampling protocol was used for all the three selected glaciers. A detailed description of the sampling procedure is described in Li et al., 2016a. Quartz filters were used to measure the mass concentration of BC, OC, and dust in the collected samples. BC and OC present in snow samples were analyzed by a filter-based thermal-optical analysis method using DRI® Model 2005 (Chow et al., 1993). Filters were analyzed at the State Key Laboratory of Cryosphere Science, Northwest Institute of Eco-Environment and Resources, Chinese Academy of Sciences. Before starting the analysis, a piece of the sampled filters was put in an oven for a few minutes to eliminate the water vapor content and volatile organic compounds. Further detailed information on the instrument and analysis method can be referred to in earlier studies (Gul et al., 2018).

Photographs of selected glaciers during the snow sampling timing

[Figure]

Figure S1. The surface condition of selected glaciers (a). Yala glacier in Nepal, (b). Thana glacier in Bhutan (c). Sachin glacier in Pakistan.

Comment#3

I want to commend the authors for their careful analysis of the discrepancy between the magnitude of BC loading in snow as reported here, compared to other regions -- on the order of 100-1000. I have just a few items I would like clarified about this.

- The authors allude to a possible cause of this discrepancy as "strong melting of surface snow and ice in the glacier ablation zone [that] could lead to BC enrichment which causes high BC concentrations (Li et al., 2017)". But how is that consistent with the description of the samples as "fresh" snow (see my comment #2)? [I know that the authors are aware of these considerations: a recent paper in Earth-Science Reviews co-authored by Kang, states that BC concentrations on an "intensely ablated surface" can be on the order of 1000 ng/g.]

- Use of WRF-Chem runs to explain these numbers is a really great idea, but I feel that characterizing the discrepancy between those results and observations (in the abstract) as "a relatively smaller magnitude" understates it, as does the phrase "almost similar" in line

330. In contrast, looking at Figure 2, the Sachin May results, I'd say the observations are ~5x the WRF-Chem results. Also, and I guess more fundamentally, does the WRM-Chem model incorporate impurity enhancement due to snow ablation?

- I appreciate the discussion of uncertainties in the paragraph beginning on line 297. However, numbers reported elsewhere in the manuscript have unreasonably high precision, in my opinion. For example, in Line 210, the authors report an average concentration of BC at Sachin, during May, to six significant figures. The authors should justify the number of significant figures in values they report, and revise accordingly.

Response #3

- Strong melting of surface snow could lead to BC enrichment causing high BC concentrations is one of the possible reasons. Besides this, we have mentioned other possible reasons (lines 221-228, given below) that can cause a discrepancy between the magnitudes of BC loading in snow. Common examples are differences in snow age, precipitation rate, local emission, the elevation of sampling sites, meteorology, and BC deposition over the glacier surfaces. Post dry deposition of LAPs over the surface of the snow is an important factor. We have mentioned this difference in the revised text, such as elevation of the sampling locations (Table 1, and lines 119,120), snow age (lines 133,134), precipitation (Table 2, and lines 120-120), meteorology (Table 2), and local emission (lines 224-228) were not same for central and western Himalayan sites. The combined effect of these factors can produce a discrepancy between the magnitudes of BC loading in snow. We also modified the snow sampling section (please have a look at response#2).

  "The BC mass concentration difference might be due to the difference in snow type, precipitation rate, and local emission, the elevation of sampling sites, meteorology, and BC deposition over the glacier surfaces. Post dry deposition of LAPs over the surface of the snow was an important factor."

  The pollutants source regions for the central and western Himalayas are different. In the case of central Himalayas, pollutants emitted during pre-monsoon convection and multiple forest fires events are effectively lifting and transported toward central Himalayan glaciers. Due to strong inversion in winter, most of the pollutants get stuck near the surface whereas in monsoon pollutants get scavenged by rain. Thus pre-monsoon is a very significant period in the transport process in the central Himalayas.

- Compared to the observations, model results are relatively smaller magnitude (mentioned in lines 249-252).
  The WRF-Chem model implicitly accounts for the surface impurity enrichment during snow ablation by using a low meltwater scavenging efficiency for BC. However, we notice that this meltwater scavenging efficiency parameter could be associated with large uncertainties (Qian et al., 2014) due to the lack of direct observational constraints (lines 253-255).

  Lines 249-252
  The discrepancies between model results and observations are due to model uncertainties from (1) the relatively coarse grid spacing that may not capture the transport over the complex TP terrain, (2) the underestimated anthropogenic emissions that are not representative of the measurement periods, and (3) deficiencies in model physical parameterizations that affects BC transport and deposition.

Lines 253-255 (we have inserted the below text in the revised manuscript)
The WRF-Chem model implicitly accounts for the surface impurity enrichment during snow ablation by using a low meltwater scavenging efficiency for BC. However, we notice that this meltwater scavenging efficiency parameter could be associated with large uncertainties (Qian et al., 2014) due to the lack of direct observational constraints.

- Thanks for pointing it out. We modified/revised the text accordingly, (given below)

Lines 205-219
The average mass concentration of LAPs in surface snow of the Yala glacier was 358 ng g$^{-1}$ for BC, 904 ng g$^{-1}$ for OC, and 22 µg g$^{-1}$ for dust in spring (May), and was relatively lower concentrations of 69 ng g$^{-1}$ for BC, 177 ng g$^{-1}$ for OC and 4 ng g$^{-1}$ for dust during autumn (September). These mass concentrations of BC and OC in surface snow were comparable to the study result conducted on the Yala glacier in May 2017 (Gul et al., 2021).
High LAP concentration in the pre-monsoon is due to an effective transport mechanism from the Indian subcontinent and an additional source such as forest fires(Kang et al., 2019; Gul et al., 2021); The average surface concentrations of BC, OC, and dust in the Thana glacier samples during the autumn season were 39 ng g$^{-1}$, 115ng g$^{-1}$ and 34 µg g$^{-1}$, respectively. Possible reasons for the lower concentration at the Thana glacier may be due to the relatively high elevation of the sampling location and relatively fresh snow. A strong effect of LAPs (BC and dust) has been observed at lower elevations in comparison to higher elevations (Li et al., 2017). The average concentration of BC, OC, and dust measured in the selected western Himalayan glacier (Sachin) during May were 2381 ng g$^{-1}$, 3896 ng g$^{-1}$ and 101 µg g$^{-1}$, respectively, and were relatively higher during October with values of 5314 ng g$^{-1}$ for BC, and 546 µg g$^{-1}$ for dust (Gul et al., 2018).

Lines 28-31
The average mass concentrations (BC 2381 ng g$^{-1}$; OC 3896 ng g$^{-1}$; dust 101 µg g$^{-1}$) in the western Himalaya (Sachin glacier) were quite high compared to the mass concentrations (BC 358 ng g$^{-1}$, OC 904 ng g$^{-1}$, dust 22 µg g$^{-1}$) at the central Himalaya (Yala glacier).

Lines 334-337
The average mass concentration of LAPs in the samples collected from the Sachin, Yala, and Thana glaciers were in the range (835 ng g$^{-1}$ to 3545 ng g$^{-1}$ for BC and 35 µg g$^{-1}$ to 253 µg g$^{-1}$ for Dust), (23 ng g$^{-1}$ to 2529 ng g$^{-1}$for BC and 1.5 µg g$^{-1}$ to 196 µg g$^{-1}$for Dust), and (21 ng g$^{-1}$ to 127 ng g$^{-1}$ for BC and 1.5 µg g$^{-1}$ to 67 µg g$^{-1}$ for Dust) respectively.

Comment#4

Figure 3 does not seem to be referred to in the body of the manuscript.

Response #4
In the revised manuscript we have referred Figure 3 at line 239, as given below.
BC and OC concentration on our selected glaciers with a comparison to other glaciers of TP and the surrounding region are shown in Fig. 3 and Table S3.

**Technical corrections**

Comment#5

Line 1: Why is "Light" capitalized in the title?

Response #5
It was by mistake and corrected the title accordingly.
"Measurement of light absorbing particles in surface snow of central and western Himalayan glaciers: spatial variability, radiative impacts, and potential source regions"

Comment#6

Line 29, Abstract: "were quite higher compared" => "were quite high compared"

Response #6
The sentence has been modified accordingly (lines 28, 29), as given below
"The average mass concentrations (BC 2381ng g$^{-1}$; OC 3896 ng g$^{-1}$; dust 101µg g$^{-1}$) in the western Himalaya (Sachin glacier) were quite high compared to the mass concentrations…"

Comment#7

Line 68, Introduction: "Mountain glaciers are the most important freshwater resources to the lives of arid and semi-arid regions" => "Mountain glaciers are the most important freshwater resources to the inhabitants of arid and semi-arid regions"

Response #7
As suggested, we have modified the sentence accordingly (lines 68, 69) as given below
Mountain glaciers are the most important freshwater resources to the inhabitants of arid and semi-arid regions.

Comment#8

Line 69, Introduction: What is "The great Himalayas"?

Response #8
The great Himalayas also called higher Himalayas or great Himalaya Range, highest and northernmost section of the Himalayan mountain ranges. Several authors in the past have used this name (Basistha et al., 2008; Nagaoka, 1990; Singh et al., 1997; Zgorzelski, 2004). However, we are agreeing to use the Himalayas instead of the great Himalayas (line 69).
"The Himalayas is considered as world's largest freshwater reservoir outside the Polar Regions (Immerzeel et al., 2010; Marcovecchio et al., 2021)."

Comment#9

Line 78, Introduction: "it is still large uncertainties for" => "large uncertainties remain regarding"

Response #9
Agree, we have modified the sentence (lines 78) as given below
"However, large uncertainties remain regarding glacier retreat driven predominately by the deposition of BC and other LAPs"

Comment#10

Line 113: I think "mostly covered by **firm**/snow" is intended to be "mostly covered by **firn**/snow".

Response #10
Agree, we have changed the sentence accordingly (line 113) as given below
"The glacier is located away from the residential area and is mostly covered by firn/snow, especially during the winter season"

Comment#11

Line 119: Punctuation issues on this line, and other places in this paragraph, need fixing.

Response #11
We modified the sentence (lines 119, 120) as given below
"In general, the Sachin is a low elevation and relatively debris-covered glacier compared to the central Himalayan glaciers (Yala and Thana)."

Comment#12

Line 131: I think "few snow samples were also collected" should be "a few snow samples were also collected".

Response #12
Agree, we have modified the sentence accordingly (lines 132, 133) as given below
"selected glaciers; however, a few snow samples were also collected from the surrounding nearby areas of the Yala and Sachin glaciers."

Comment#13

Line 159: I don't understand the sentence: "RF-based on measured BC and dust concentration in our samples were estimated using the following equation." I understand that "RF" means "Radiative Forcing" … but what is "RF-based"? Also, the formatting of the equation on the following line is unconventional.

Response #13
Agree, we have modified the sentence and equation (line 164, 165) as given below
RF for the snow samples was estimated by following Eq. (1):
$$RF_x = R_{in-short} * \triangle \alpha_x \quad (1)$$

Comment#14

Line 169: The sentence "To identify the potential source region of pollution for the central and western Himalayan glaciers, the weather research and forecasting (WRF) model coupled with chemistry (WRF-Chem version 3.9.1.1) (Grell et al., 2005) tagged-tracer simulations for the selected sites" seems to be missing a verb.

Response #14
We have modified the sentence (175-177) as given below
"To identify the potential source region of pollution arriving at the observation sites, we used the weather research and forecasting (WRF) model coupled with chemistry (WRF-Chem version 3.9.1.1) simulations (Grell et al., 2005). The model uses region-tagged atmospheric BC tracers for different regions across the world."

Comment#15

Line 207: "Possible reasons for the lowest concentration" => "Possible reasons for the lower concentration" (I think).

Response #15
Agree, we have modified the sentence accordingly (line 215) as given below
"Possible reasons for the lower concentration at the Thana glacier may be due to the relatively high elevation of sampling location and relatively fresh snow."

Comment#16

Line 389: The Gul et al citation does not seem to include the year of publication.

Response #16
We have included the year of publication (line 406) as given below
Gul, C., Mahapatra, P.S., Kang, S., Singh, P.K., Wu, X., He, C., Kumar, R., Rai, M., Xu, Y., Puppala, S.P., Black carbon concentration in the central Himalayas: impact on glacier melt and potential source contribution, Environmental Pollution, https://doi.org/10.1016/j.envpol.2021.116544, 2021

Comment#17

Line 558, Figure 2: I don't understand the simulation results for Sachin(Oct) – maybe the figure was garbled? Also, the caption doesn't explain the meaning of the "x" and "+" symbols.

Response #17
We have slightly modified the figure (lines 558-561) given below. In the caption of the figure, we add a short sentence "Stars (*) are representing outliers".

[Figure]

**Fig. 2.** **Whisker plots of black carbon (red box) and organic carbon (black box) concentrations (ng/g) in snow samples were collected from three different glaciers in the spring and autumn of 2016. The yellow boxes are representing BC content in surface snow from WRF-Chem simulations. Stars (*) are representing outliers.**

**References**

Basistha, Ashoke, D. S. Arya, and N. K. Goel. "Spatial distribution of rainfall in Indian Himalayas–a case study of Uttarakhand region." Water Resources Management 22.10 (2008): 1325-1346.

Dang C, Fu Q, Warren SG (2016) Effect of snow grain shape on snow albedo. J Atmos Sci 73(9):3573–3583

He, C., Flanner, M. G., Chen, F., Barlage, M., Liou, K.-N., Kang, S., Ming, J., and Qian, Y.: Black carbon-induced snow albedo reduction over the Tibetan Plateau: uncertainties from snow grain shape and aerosol–snow mixing state based on an updated SNICAR model, Atmos. Chem. Phys., 18, 11507–11527, https://doi.org/10.5194/acp-18-11507-2018, 2018

Jamieson, J. B., and J. Schweizer, Texture and strength changes of buried surface hoar layers with implications for dry snow-slab avalanche release, J. Glaciol., 46(152), 151– 160, 2000

Qian, Y., Wang, H., Zhang, R., Flanner, M. G., & Rasch, P. J. (2014). A sensitivity study on modeling black carbon in snow and its radiative forcing over the Arctic and Northern China. Environmental Research Letters, 9(6), 064001.

Li, C., Bosch, C., Kang, S., Andersson, A., Chen, P., Zhang, Q.,Cong, Z., Chen, B., Qin, D., and Gustafsson, Ö.: Source of blackcarbon to the Himalayan-Tibetan Plateau glaciers, Nat. Com-mun., 7, 12574, https://doi.org/10.1038/ncomms12574, 2016a.

Nagaoka, Shinji. "The glacial landforms in the Manang Valley, north of the great Himalayas, central Nepal." (1990).

Singh, G. S., S. C. Ram, and J. C. Kuniyal. "Changing traditional land use patterns in the Great Himalayas: a case study of Lahaul Valley." Journal of Environmental Systems 25 (1997): 195-211.

Zgorzelski, Marek. "Orographic Barrier of the Great Himalayas." Miscellanea Geographica 11.1 (2004): 71-74.

----------------------------- End of reviewer's 1 comments ----------------------------

**Reviewer #2:**

On behalf of all the authors, we would like to convey our gratitude to the editor and the reviewers for considering the present work.